# The Spatio-Temporal Variations of GPP and Its Climatic Driving Factors in the Yangtze River Basin during 2000–2018

**Chong Nie** [1,2]**, Xingan Chen** [3]**, Rui Xu** [1,2]**, Yanzhong Zhu** [1,2]**, Chenning Deng** [1] **and Queping Yang** [1,2,]*****

1  Chinese Research Academy of Environmental Sciences, Beijing 100012, China; nie.chong@craes.org.cn (C.N.); xurui198877@126.com (R.X.); zhu_yanzhong@craes.org.cn (Y.Z.); dengcn@mail.bnu.edu.cn (C.D.)
2  National Joint Research Center for Yangtze River Conservation, Beijing 100012, China
3  State Key Laboratory of Hydroscience and Engineering, Department of Hydraulic Engineering, Tsinghua University, Beijing 100084, China; cxa19@mails.tsinghua.edu.cn
*  Correspondence: yangqp@craes.org.cn

**Abstract:** Terrestrial gross primary productivity (GPP) is the major carbon input to the terrestrial ecosystem. The Yangtze River Basin (YRB) holds a key role in shaping China's economic and social progress, as well as in ecological and environmental protection. However, how the GPP in the YRB responds to the climate factors remain unclear. In this research, we applied the Vegetation Photosynthesis Model (VPM) GPP data to explore the spatial and temporal variations of GPP in the YRB during 2000–2018. Based on the China Meteorological Forcing Dataset (CMFD), the partial least squares regression (PLSR) method was employed to identify the GPP responses to changes in precipitation, temperature, and shortwave radiation between 2000 and 2018. The findings showed that the long-term average of GPP in the YRB was $1153.5 \pm 472.4$ g C m$^{-2}$ yr$^{-1}$ between 2000 and 2018. The GPP of the Han River Basin, the Yibin-Yichang section of the Yangtze River mainstream, and the Poyang Lake Basin were relatively high, while the GPP of the Jinsha River Basin above Shigu and the Taihu Lake Basin were relatively low. A significant upward trend in GPP was observed over the 19-year period, with an annual increase rate of 8.86 g C m$^{-2}$ yr$^{-1}$ per year. The GPP of the Poyang Lake Basin and Jialing River Basin grew much faster than other water resource regions. Savannas and forests also had relatively higher GPP rate of increase compared to other vegetation types. The relative contributions of precipitation, temperature, and shortwave radiation to GPP variations in the YRB were $13.85 \pm 13.86\%$, $58.87 \pm 9.79\%$, and $27.07 \pm 15.92\%$, respectively. Our results indicated that temperature was the main climatic driver on the changes of GPP in the YRB. This study contributes to an in-depth understanding of the variations and climate-impacting factors of vegetation productivity in the YRB.

**Keywords:** terrestrial gross primary productivity (GPP); Yangtze River Basin; climate change



## 1. Introduction

To keep human-induced global warming below 2 °C, it is essential to rapidly decrease $CO_2$ emissions and implement measures to increase the accumulation and sequestration of $CO_2$ [1,2]. Thus, improving the terrestrial ecosystem carbon sink is a significant approach to decelerate the ongoing rise in atmospheric $CO_2$ concentrations and to achieve the carbon neutrality target [2,3]. Terrestrial gross primary productivity (GPP) is defined as the quantity of $CO_2$ that land plants 'fix' into organic material via photosynthesis. It stands as the primary carbon input to the terrestrial ecosystem [4–6]. GPP holds significance for human welfare as it forms the foundation to produce food, fiber, and wood [7].

The variations of GPP can be attributed to climate changes and anthropogenic activities. It should be noted that studying the influences of climate changes on GPP is a crucial first step [8]. First, climate variables like temperature, precipitation, and solar radiation directly influence GPP [9]. Moreover, those effects may be more profound and potentially

alter the structure of terrestrial ecosystems as well as the physiology of vegetation [10,11]. Understanding the climate sensitives of GPP helps predict how ecosystems might respond to future climate changes, whether they are natural or anthropogenically amplified. It plays an important role for us in formulating measures to address climate changes. Second, investigating the influences of climate changes on GPP is the basis of partitioning the influences of anthropogenic and naturally induced changes in GPP. Based on the understanding of the contributions of climate changes on GPP, the additional or synergistic impacts of anthropogenic factors, such as land-use change, pollution, or direct human interference can be easily measured [12,13]. Thus, in this study, we mainly focused on the GPP variations and its responses to climate factors.

Even though there were many studies investigating GPP [14], it remains challenging to monitor GPP across different scales [15]. Over the past decades, eddy covariance (EC) technologies have become crucial tools for deriving GPP through direct observations of $CO_2$ exchange at the canopy–atmosphere interface [16]. However, the EC data is limited from 100 m to 2000 m surrounding the flux site [17] and is thus usually constrained by scale and expenses, which is not suitable for spatial analysis [18]. Another commonly used method for estimating GPP involves the integration of satellite-based estimates and ecosystem models [19,20]. Satellite remote sensing technologies enable the rapid and cost-effective acquisition of extensive land surface data that reflects vegetation-related and environmental variables [21]. Over the past few decades, multiple GPP models using remote sensing data have been developed to model GPP on a regional and global scale [4,22–25]. Nevertheless, different products based on different models may have uncertainties.

There exists 4 advanced global GPP products with exceptional spatial and temporal resolutions, which are the Moderate Resolution Imaging Spectroradiometer Photosynthesis (MOD17) [26], the Vegetation Photosynthesis Model (VPM) [27], the Breathing Earth System Simulator (BESS) [28], and the Penman–Monteith–Leuning (PML) models [29]. MOD17 and VPM GPP data were generated based on light use efficiency (LUE) models [30]. BESS and PML are process-based models. MOD17 and PML employed vapor pressure deficit (VPD) as an indicator of moisture stress [29,31–33], and BESS applied relative humidity (RH) via the Ball–Berry model [34,35] to consider the water stress conditions. Both VPD and RH are parameters associated with atmosphere moisture [36]. VPM applied a satellite-based land surface water index (LSWI) to consider water stress conditions, which is closely associated with the moisture status of plants [27,37,38]. Pei et al. [39] found that these four products demonstrated satisfactory accuracies during non-drought years when using EC data as a reference, but the VPM GPP products performed much better compared with the other three products under drought years. Therefore, the VPM GPP products outperformed the other three GPP products when investigating the responses of GPP to variations in environmental moisture changes.

The Yangtze River Basin (YRB) holds significant importance as a crucial industrial and agricultural production zone, serving as an ecological security barrier in China. As a unique and complete natural ecosystem, the YRB has great ecological functions such as water and soil conservation, and holds a pivotal role in maintain the ecological balance and security of both the surrounding regions and the entire country. Efforts had been made for scholars to investigate the dynamic changes of vegetation and its productivity in the YRB. Qu et al. [8] revealed that the Normal Difference Vegetation Index (NDVI) exhibited a growing trend in the growing season from 1982 to 2015 in the YRB, which was mainly attributed to the rising temperature. Peng, et al. [40] found that temperature can significantly affect vegetation changes in Sichuan, which comprised 25% of the entire YRB area. Based on different scenarios, Liu et al. [41] found that land use alterations resulted in a favorable impact on total production by 3.42 Tg C, and the climate changes contributed to a negative effect by −1.44 Tg C. Since different data sources have different results, using more reliable data and methods can better capture the GPP changes and quantify its driving forces. Thus, using the VPM GPP data may prove to be a more precise estimation of the

GPP trend and facilitate a more effective assessment of the relative contributions of climate changes in the YRB.

In this study, we first evaluated the spatio–temporal variations of the GPP in the YRB based on the VPM GPP data. Then, the responses of GPP to the variations of climate factors were explored. In addition, the contributions of those driving factors were quantified based on statistical models.

## 2. Materials and Methods

### 2.1. Study Area

The Yangtze River is the largest river in China, with a total length of nearly 6400 km. The YRB ($24°30'$–$35°45'$ N and $90°33'$–$122°25'$ E) extends from the eastern part of the Tibetan Plateau to the East China Sea, encompassing an area of nearly $1.8 \times 10^6$ km$^2$, which constitutes approximately 18.8% of China's total land area (Figure 1a). The YRB is characterized by diverse topographies, intricate river networks, and numerous tributaries. The YRB can be divided into 12 water resource regions according to the national water resources management zoning, which are the Jinsha River Basin above Shigu (JSJ-1), the Jinsha River Basin below Shigu (JSJ-2), the Mintuo River Basin (MTJ), the Jialing River Basin (JLJ), the Wu River Basin (WJ), the Yibin-Yichang section of the Yangtze River mainstream (UM), the Dongting Lake Basin (DTL), the Han River Basin (HJ), the Poyang Lake Basin (PYL), the Yichang-Hukou section of the Yangtze River mainstream (MM), the Yangtze River mainstream below Hukou (LM), and the Taihu Lake Basin (TL) (Figure 1a).

The YRB exhibits a typical subtropical monsoon climate, with a decline in both temperature and precipitation from the southwest to the northwest. The average annual temperature varies from 12.6–28.3 °C, and the annual precipitation varies between 500 to 2500 mm [42–44]. The distributions of precipitation, temperature, and shortwave radiation of the YRB during 2000–2018 were shown in Figure 1b–d. It can be observed that precipitation and temperature in the YRB both exhibited decreasing patterns from the upper reaches to the lower reaches (from northwest to southeast). This is particularly evident in the lower reaches of the Yangtze River, where precipitation is often greater than 500 mm. However, the shortwave radiation showed the opposite pattern, with a decreasing trend from the upper reaches to the lower reaches.

The YRB boasts abundant vegetation resources, with forest reserves constituting almost 25% of the entire country's forested areas. The source region and upper reaches of the YRB are mainly characterized by alpine meadows and natural grasslands. The middle reaches of the YRB are primarily covered by deciduous broad-leaved forests and evergreen broad-leaved forests. Forests and savannas are primarily distributed in regions with favorable water and heat conditions. Cropland is extensively distributed across the middle and downstream plain regions (Figure 1e). Over the past decades, the vegetation cover in the YRB has experienced significant changes. Initiatives such as ecological afforestation projects, notably the Grain for Green Program, have facilitated the shift from agricultural land to forested areas and grasslands [8]. However, the dense population and industrial presence along the Yangtze River, particularly in the Yangtze River Delta (YRD), present a challenge to the available space for vegetation expansion.

### 2.2. Data Sources

We applied the VPM GPP dataset in this study, which is from simulations of the satellite-based Vegetation Photosynthesis Model (VPM) [27]. This dataset has been updated to December 2019, with the spatial resolution of $0.05° \times 0.05°$. It can be acquired from http://data.tpdc.ac.cn/en/data/582663f5-3be7-4f26-bc45-b56a3c4fc3b7/ (accessed on 11 August 2023). The VPM GPP dataset was generated using an enhanced light use efficiency theory, forcing by the moderate resolution imaging spectroradiometer (MODIS) data, and reanalysis climate dataset. Furthermore, it utilizes an advanced vegetation index (VI) gap-filling and smoothing algorithm, along with a partition approach for C3/C4 photosynthesis pathways [27,45].

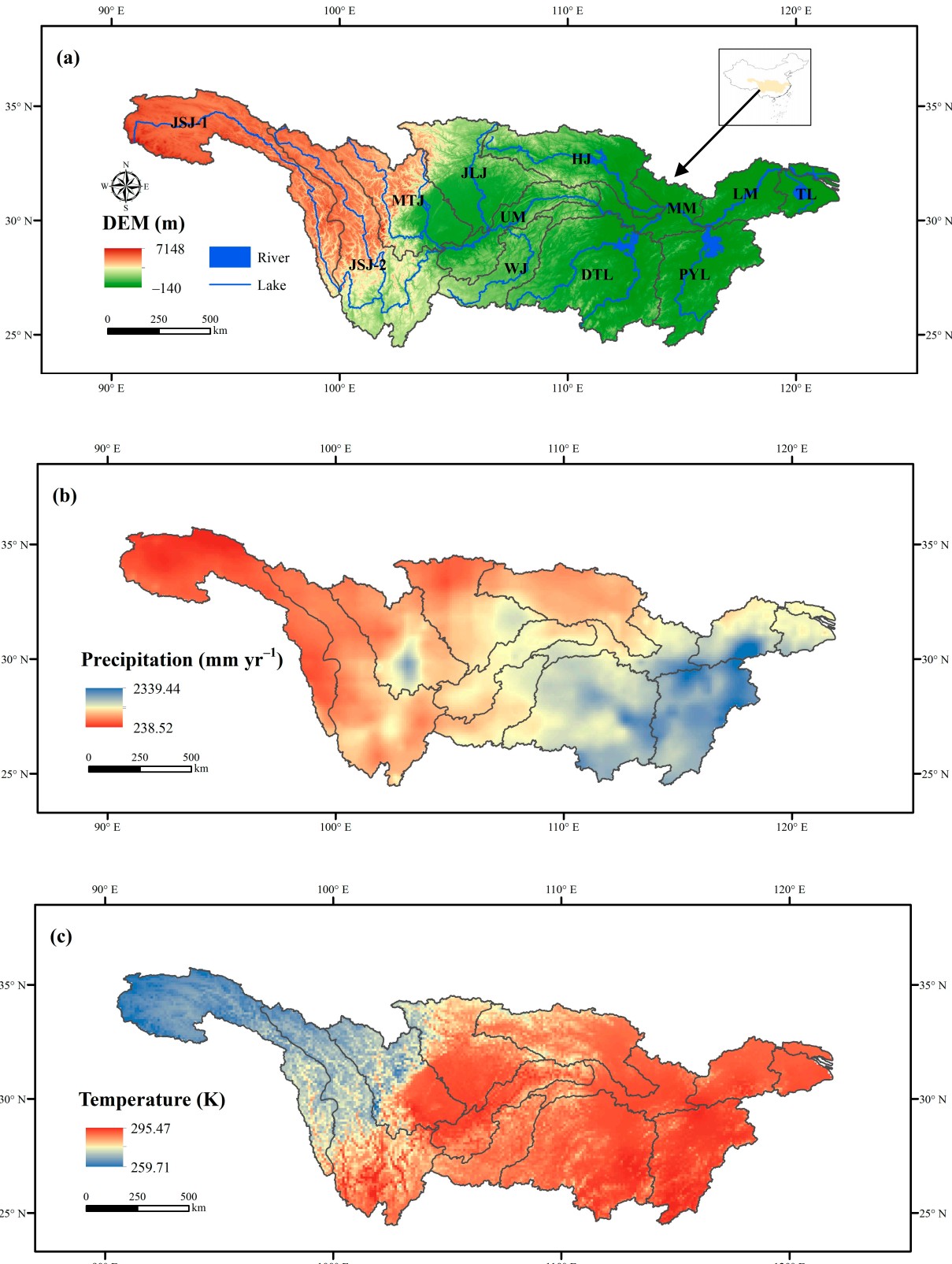

**Figure 1.** *Cont*.

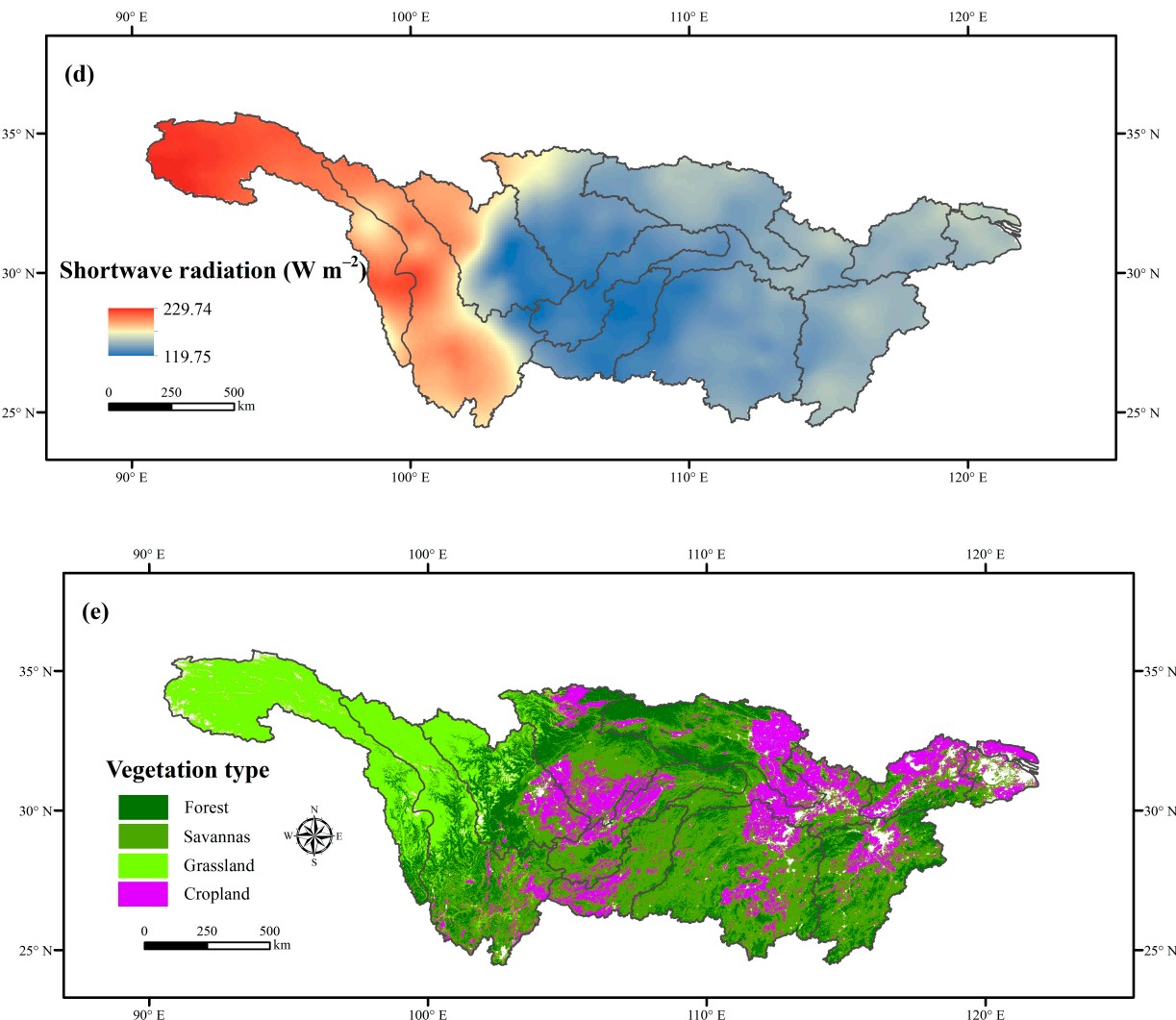

**Figure 1.** The location and 12 water resource regions of the YRB (**a**). The spatial distributions of long-term averages of precipitation (**b**), temperature (**c**), shortwave radiation (**d**), and vegetation cover (**e**) of the YRB.

In this study, we selected climate factors that may have a significant impact on vegetation productivity, such as temperature, precipitation, and shortwave radiation. Those data were derived from the China Meteorological Forcing Dataset (CMFD). CMFD offers a spatial resolution of $0.1° \times 0.1°$ and a temporal resolution of 1 day (Figure 1b–d). These data were downloaded from the "Cold and Arid Regions Science Data Center at Lanzhou" (http://westdc.westgis.ac.cn/data/7a35329c-c53f-4267-aa07-e0037d913a21 (accessed on 11 August 2023)) [46]. This dataset was produced by integrating data from various sources, encompassing instantaneous near surface air temperature, 3-h mean (ranging from −1.5 to +1.5 h), surface downward shortwave radiation, etc. The vegetation type data were derived from MODIS product (MCD12Q1) in 2010 (Figure 1e). The vegetation type categories in MCD12Q1 were based on the International Geosphere-Biosphere Programme (IGBP) classifications. As we mainly focused on the forest, savanna, grassland, and cropland types, we combined deciduous evergreen needleleaf forests, evergreen broadleaf forests, deciduous needleleaf forests, and mixed forests into the forest category. We combined woody savannas and savannas into the savannas category.

## 2.3. Data Processing and Statistical Analysis

Both VPM GPP dataset and CMFD data were aggregated to the spatial resolution of $0.1° \times 0.1°$ and the temporal resolution of 8 days. We first evaluated the spatio–temporal variations of GPP in the YRB, analyzing the changes of GPP among different water resource regions and vegetation types. The spatio–temporal variations of precipitation, temperature, and shortwave radiation were also explored. Considering the possible correlations among climate factors, we applied Partial Least Squares Regression (PLSR) to analyze the response of GPP to variations in precipitation, temperature, and shortwave radiation. PLSR is a statistical modeling method based on Principal Component Analysis (PCA) and multiple linear regression techniques. The PLSR method was commonly used in studies when the predictors are highly correlated [47,48]. We quantified the relative contributions of precipitation, temperature, and shortwave radiation to the variations of GPP.

$$\xi_p = 100\% \times \frac{|\eta_p|}{|\eta_p| + |\eta_t| + |\eta_s|} \tag{1}$$

$$\xi_t = 100\% \times \frac{|\eta_t|}{|\eta_p| + |\eta_t| + |\eta_s|} \tag{2}$$

$$\xi_s = 100\% \times \frac{|\eta_s|}{|\eta_p| + |\eta_t| + |\eta_s|} \tag{3}$$

$\xi_p$(%), $\xi_t$(%) and $\xi_s$(%) represented the relative contributions of precipitation, temperature, and shortwave radiation to the variation of GPP in the YRB, respectively. $\eta_p$, $\eta_t$, and $\eta_s$ represented the standardized sensitivity coefficients of GPP to the changes in precipitation, temperature, and shortwave radiation, respectively. Before applying the PLSR method, all variables were standardized using the z-score method. The above statistical analysis was conducted using RStudio (Version 2023.06.1+524).

## 3. Results

### 3.1. The Spatial Distributions of Annual GPP

We first analyzed the spatial distributions of annual GPP across the YRB during the period 2000–2018 (Figure 2a). The long-term average of GPP in the YRB was $1153.5 \pm 472.4$ g C m$^{-2}$ yr$^{-1}$. The total amount of GPP in the YRB was $2.08 \times 10^{15}$ g C yr$^{-1}$. The multi-year average GPP of the YRB mainly ranged from 800 to 1800 g C m$^{-2}$ yr$^{-1}$, and the regions in this interval accounted for 76.24% of the entire YRB (Figure 2b). The high-value areas of GPP (higher than 1500 g C m$^{-2}$ yr$^{-1}$) covered 25.77% of the YRB and were primarily situated in the middle and lower reaches. The low GPP (lower than 500 g C m$^{-2}$ yr$^{-1}$) areas covered 13.04% of the YRB and were mainly distributed in upper reaches of the YRB (Qinghai Province). The annual average GPP showed a significantly decreasing trend with the increase of latitude (Figure 2c), and the rate of change in GPP varied with increasing latitude. Results show that below 32.125° N, GPP decreased slightly with the increase in latitude (*Slope* = $-47.83$ g C m$^{-2}$ yr$^{-1}$ degree$^{-1}$, $p < 0.01$, $R^2 = 0.87$), but above 32.125°, GPP decreased sharply (*Slope* = $-419.18$ g C m$^{-2}$ yr$^{-1}$ degree$^{-1}$, $p < 0.01$, $R^2 = 0.94$). In general, GPP would be reduced by 107.31 g C m$^{-2}$ yr$^{-1}$ for every 1 degree increase in latitude in the YRB ($p < 0.01$, $R^2 = 0.71$). The annual average GPP exhibited a pattern of initially rising and subsequently declining with the increase in longitude. The increasing trend (*Slope* = 96.78 g C m$^{-2}$ yr$^{-1}$ degree$^{-1}$, $p < 0.01$, $R^2 = 0.96$, Figure 2d) was more rapidly than the decreasing trend (*Slope* = $-45.28$ g C m$^{-2}$ yr$^{-1}$ degree$^{-1}$, $p < 0.01$, $R^2 = 0.61$). The maximum GPP value was 2076 g C m$^{-2}$ yr$^{-1}$ and was mainly distributed around 110° E.

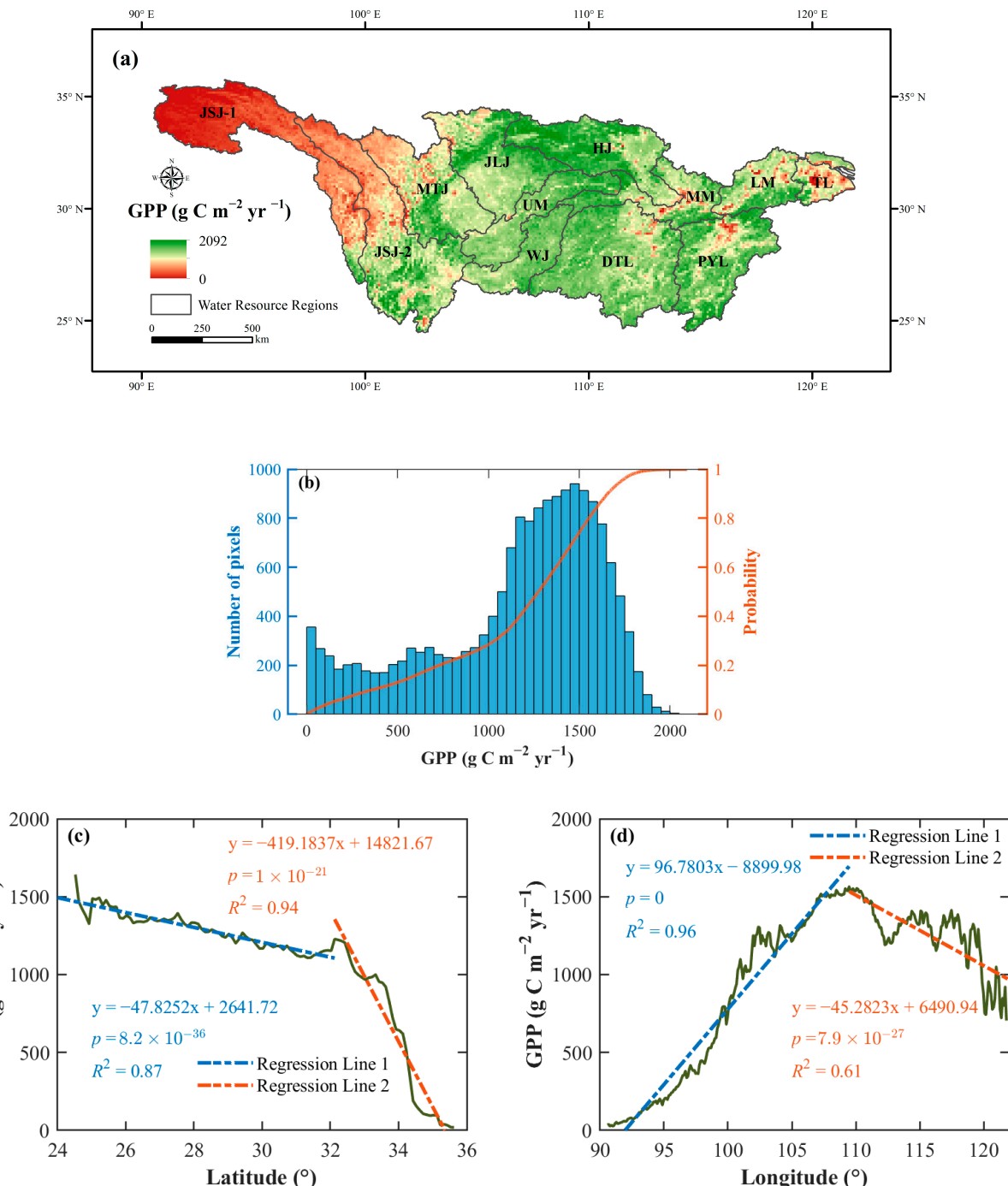

**Figure 2.** The spatial variations of annual average GPP in the YRB between 2000 and 2018 (**a**) and its statistical characteristics (**b**). The variations of the long-term average of GPP with the changes in latitude (**c**) and longitude (**d**).

We investigated the variations of the annual GPP in different water resource regions and vegetation types in the YRB, the results of which were shown in Figure 3. Similar to the above results, the annual average GPP of 12 water resource regions first showed an increase and then a decrease with the increase in longitude. In general, the Han River Basin (HJ) had the largest annual average GPP of $1502.3 \pm 273.1$ g C m$^{-2}$ yr$^{-1}$, followed by the Yibin-Yichang section of the Yangtze River mainstream (UM) and the Poyang Lake Basin (PYL), with an annual average GPP of $1401.5 \pm 184.0$ g C m$^{-2}$ yr$^{-1}$ and $1382.9 \pm 304.1$ g C m$^{-2}$ yr$^{-1}$, respectively. The Jinsha River Basin above Shigu (JSJ-1) had

the lowest average GPP of $321.8 \pm 328.1$ g C m$^{-2}$ yr$^{-1}$, followed by the Taihu Lake Basin (TL) with $891.7 \pm 404.8$ g C m$^{-2}$ yr$^{-1}$. In addition, the Jinsha River Basin below Shigu (JSJ-2) had the highest standard deviation of 428.5 g C m$^{-2}$ yr$^{-1}$ followed by the Taihu Lake Basin (TL) of 404.8 g C m$^{-2}$ yr$^{-1}$.

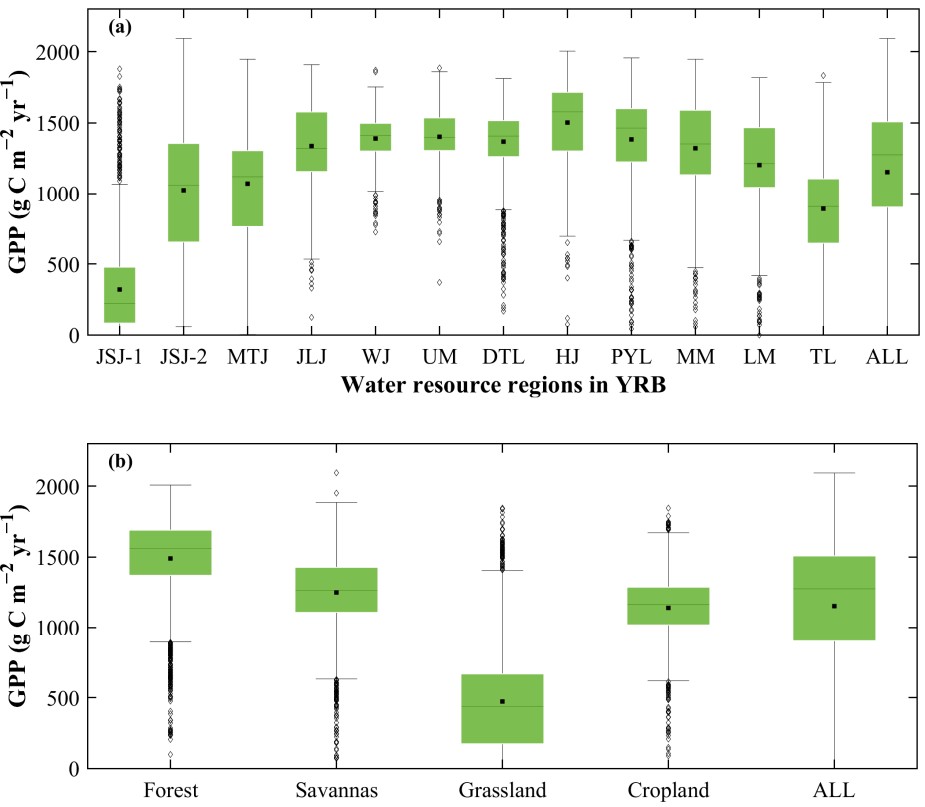

**Figure 3.** Boxplot of the variations of GPP in different water resource regions (**a**), and vegetation types (**b**) in the YRB. Boxplot elements: square points = mean values; horizontal lines within boxes = medians; whiskers = 1.5 times of interquartile ranges; and points outside the boxes = outliers.

We analyzed the GPP variations in different vegetation types (Figure 3b). Results showed that forests had much higher GPP values, with an average of $1476.9 \pm 287.3$ g C m$^{-2}$ yr$^{-1}$, followed by savannas and cropland, with GPP averages of $1345.7 \pm 274.2$ g C m$^{-2}$ yr$^{-1}$ and $1130.4 \pm 246.2$ g C m$^{-2}$ yr$^{-1}$, respectively. Grassland had much lower GPP values with an average of $469.0 \pm 354.3$ g C m$^{-2}$ yr$^{-1}$.

### 3.2. The Spatio–Temporal Variations of GPP and Climate Factors in the YRB

We analyzed the overall changes of the GPP in the YRB spanning the years 2000 to 2018, which were depicted in Figure 4. The annual average GPP in the YRB fluctuated from 999.77 to 1275.60 g C m$^{-2}$ yr$^{-1}$, and the GPP reached the highest value in 2013. The annual average GPP increased significantly during the 19-year period ($R^2 = 0.54$, $p < 0.01$), with an annual average GPP increase rate of 8.86 g C m$^{-2}$ yr$^{-1}$ per year. Figure 5a–b illustrated the spatio–temporal variations in the GPP change rate across the YRB during 2000–2018. Results showed that the change rate of GPP in the YRB mainly ranged from $-26.55$ to 36.99 g C m$^{-2}$ yr$^{-2}$, with an average of $9.01 \pm 8.31$ g C m$^{-2}$ yr$^{-2}$. A total of 68.52% of the regions were significant at the significance level of $p < 0.1$, and the average $R^2$ was $0.27 \pm 0.25$. In addition, 87.74% of the YRB regions had increasing trends of GPP with an average GPP change rate of 10.80 g C m$^{-2}$ yr$^{-2}$. Among them, 73.71% of the regions were significant at the significance level of $p < 0.1$. A total of 12.26% of the YRB had decreasing trends with an average GPP change rate of $-4.71$ g C m$^{-2}$ yr$^{-2}$, but the results of regression

analysis showed that 96.15% of those regions with negative slopes were not significant ($p > 0.1$). The regions with reduced GPP were primarily situated in the lower reaches of the Yangtze River and the headwater regions.

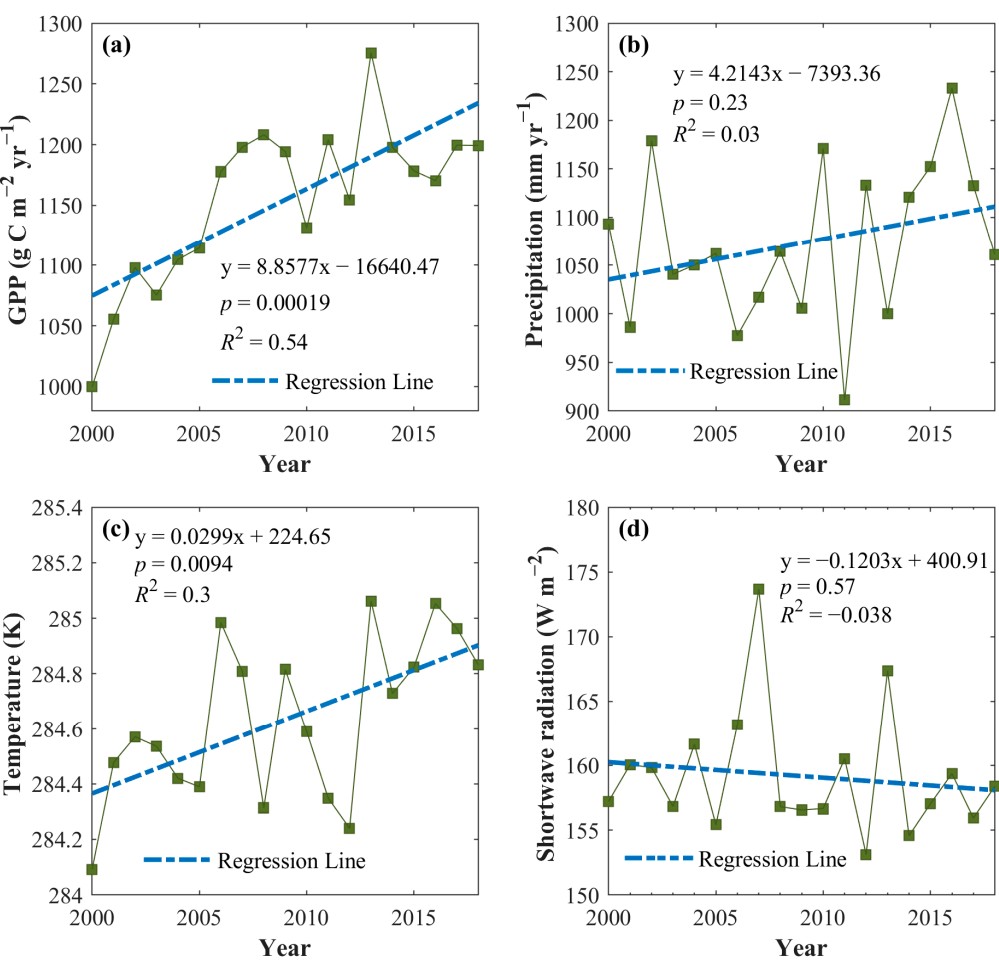

**Figure 4.** The variations of annual average GPP (**a**), precipitation (**b**), temperature (**c**), and shortwave radiation (**d**) from 2000 to 2018 in the YRB.

We analyzed the change rate of GPP across various water resource regions and vegetation types in the YRB, as illustrated in Figure 5c–d. From the upper reaches to the lower reaches of the YRB, the GPP change rates of the 12 water resource regions presented an initial upward trend followed by a subsequent decline, as per the results in Section 3.1. The change rate of GPP was mainly positive in the upper reaches and finally became negative in the estuary, as shown in Figure 5c. The Jinsha River above Shigu (JSJ-1) had the lowest positive GPP change rate, which is $2.25 \pm 4.32$ g C m$^{-2}$ yr$^{-2}$, followed by the Yangtze River mainstream below Hukou (LM), which is $3.45 \pm 7.08$ g C m$^{-2}$ yr$^{-2}$. The Poyang Lake Basin (PYL), the Jialing River Basin (JLJ), and the Han River Basin (HJ) had relatively higher GPP growth rates, with averages of $13.98 \pm 7.07$ g C m$^{-2}$ yr$^{-2}$, $13.81 \pm 7.69$ g C m$^{-2}$ yr$^{-2}$, and $13.54 \pm 7.95$ g C m$^{-2}$ yr$^{-2}$, respectively. The GPP in the Taihu Lake Basin (TL) mainly showed decreasing trends, and the change rate of GPP was $-4.22 \pm 9.85$ g C m$^{-2}$ yr$^{-2}$. Only 27.71% of the Taihu Lake Basin (TL) had increasing GPP trends. GPP in the vegetation area showed increasing trends in the YRB (Figure 5d). Overall, savannas and forests had relatively higher GPP increasing rates, with averages of $13.07 \pm 6.79$ g C m$^{-2}$ yr$^{-2}$ and $12.16 \pm 6.54$ g C m$^{-2}$ yr$^{-2}$, respectively. Grassland had much lower GPP increasing rates with an average of $4.27 \pm 4.34$ g C m$^{-2}$ yr$^{-2}$.

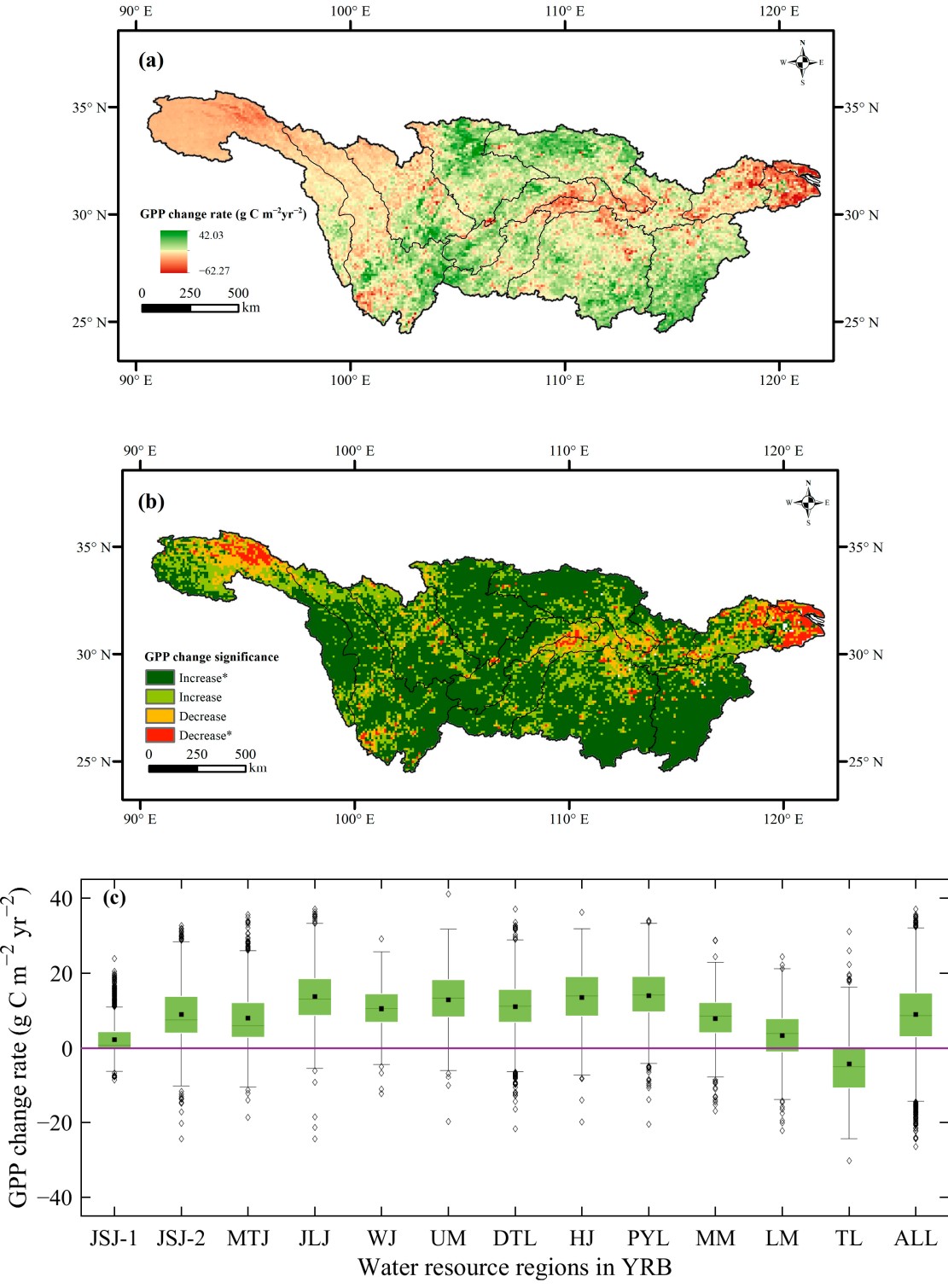

**Figure 5.** *Cont.*

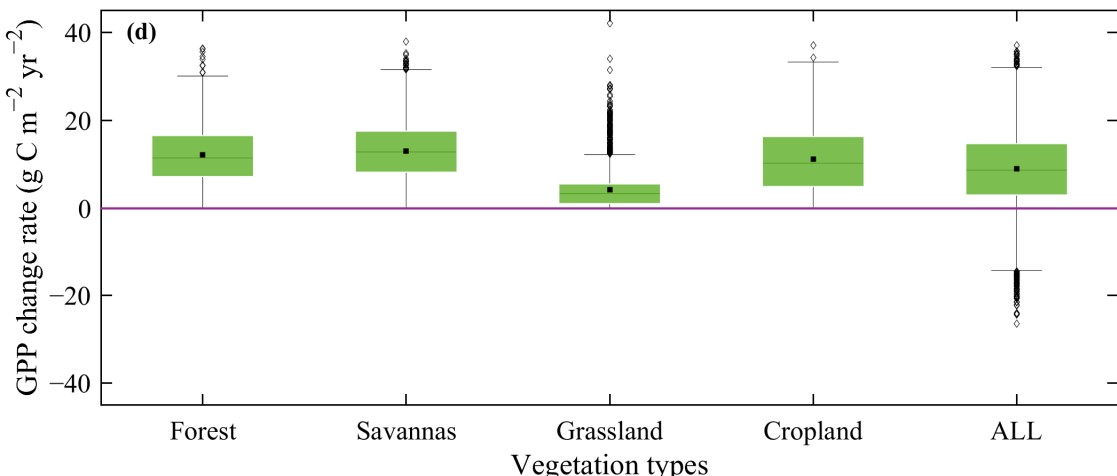

**Figure 5.** The variations of GPP change rate (**a**), the significance of regression analysis (**b**), the boxplot of the GPP change rate in 12 water resource regions (**c**), and the boxplot of GPP change rate in different vegetation types (**d**). "Increase*" represented that the increasing trends were significant ($p < 0.1$). "Increase" represented that the increasing trends were insignificant ($p > 0.1$). "Decrease*" represented that the decreasing trends were significant ($p < 0.1$). "Decrease" represented that the decreasing trends were insignificant ($p > 0.1$). Boxplot elements: square points = mean values; horizontal lines within boxes = medians; whiskers = 1.5 times of interquartile ranges; and points outside the boxes = outliers. Purple lines represented the "zero value" lines.

We also investigated the spatio–temporal changes of climate factors in the YRB, as shown in Figure 4b–d and supplementary file (Figures S1–S3). We observed an increasing trend of precipitation in the YRB, but the trend of annual averages was not significant (*Slope* = 4.21 mm yr$^{-2}$, $p > 0.1$, Figure 4b). A total of 65.19% of the regions in the YRB had increasing precipitation trends, and only 24.08% of them were significant ($p < 0.1$, Figure S1). The decreasing trends of precipitation were mainly observed in the upper reaches of the YRB such as the Jinsha River Basin above Shigu (JSJ-1) and the Jinsha River Basin below Shigu (JSJ-2). The temperature had significantly increased (*Slope* = 0.03 K yr$^{-1}$, $R^2 = 0.3$, $p < 0.05$, Figure 4c) in the YRB during 2000–2018. A total of 57.37% of the regions with an increasing temperature trend were significant (Figure S2). Results show that shortwave radiation in the YRB was reduced, but the trend was not significant (*Slope* = $-0.12$ W m$^{-2}$ yr$^{-1}$, $p > 0.1$, Figure 4d). A total of 20.74% of the regions with decreasing trends of shortwave radiation were significant (Figure S3b).

### 3.3. The Responses of Precipitation, Temperature and Shortwave Radiation to the Variations of GPP

We calculated the standardized sensitivity coefficients of precipitation ($\eta_p$), temperature ($\eta_t$), and shortwave radiation ($\eta_s$), the results of which were shown in Figure 6. The averages of $\eta_p$, $\eta_t$, and $\eta_s$ were $0.11 \pm 0.15$, $0.58 \pm 0.11$, and $0.24 \pm 0.20$, respectively. The relative contributions of precipitation ($\xi_p$), temperature ($\xi_t$), and shortwave radiation ($\xi_s$) to the changes of GPP in the YRB were shown in Figure 7. The averages of $\xi_p$, $\xi_t$ and $\xi_s$ were $13.85 \pm 13.86\%$, $58.87 \pm 9.79\%$, and $27.07 \pm 15.92\%$, respectively. The proportions of precipitation, temperature, and shortwave radiation as dominant factors influencing the variation of GPP in the YRB were 2.07%, 96.69%, and 1.24%, respectively. Among the 3 climate factors, the GPP of the YRB was most sensitive to temperature changes, followed by shortwave radiation. GPP in the YRB was less sensitive to the changes in precipitation.

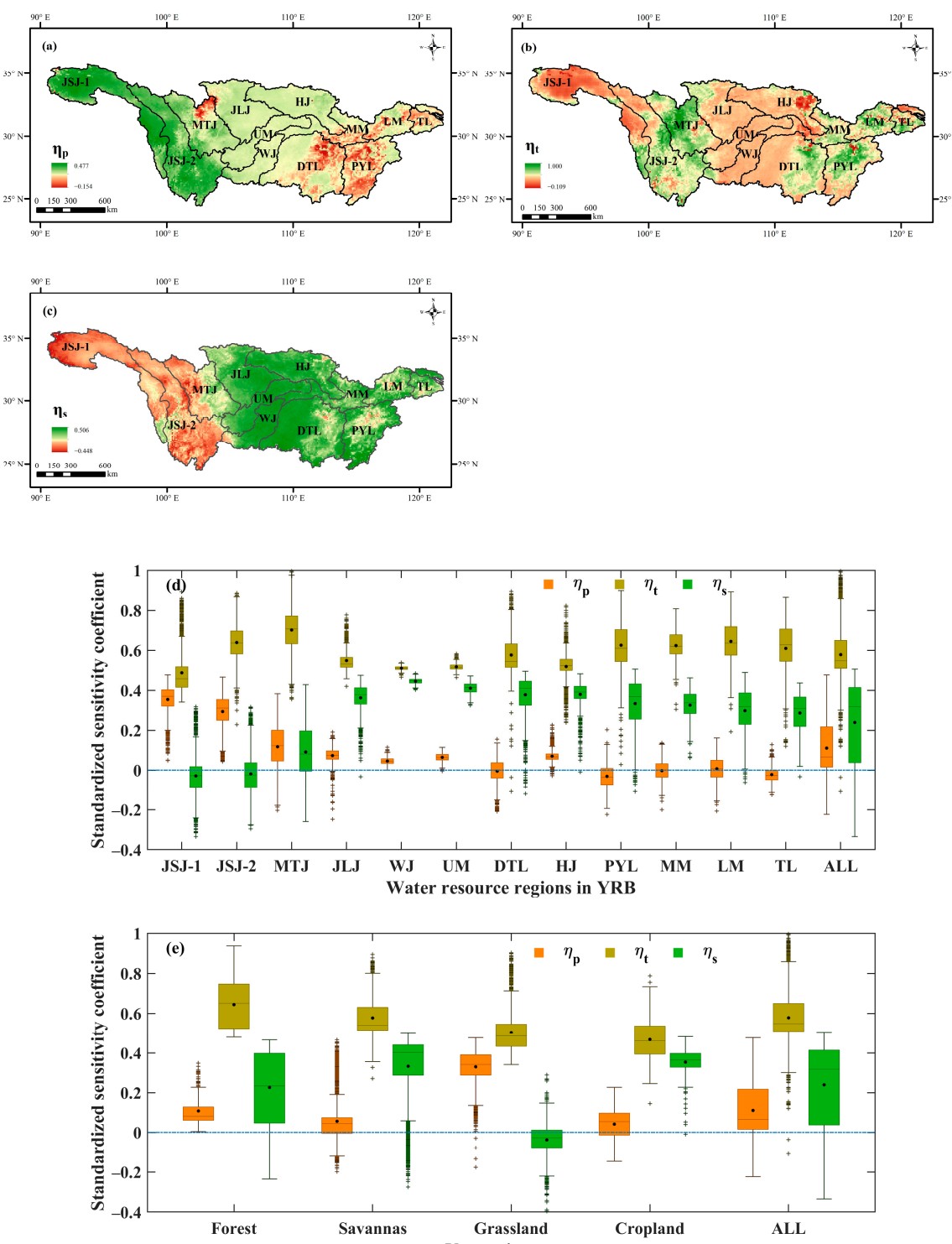

**Figure 6.** The spatial distributions of $\eta_p$ (**a**), $\eta_t$ (**b**) and $\eta_s$ (**c**) in the YRB. Boxplots of $\eta_p$, $\eta_t$, and $\eta_s$ in 12 water resource regions (**d**) and vegetation types (**e**). Boxplot elements: solid points = mean values; horizontal lines within boxes = medians; whiskers = 1.5 times of interquartile ranges; and "+" outside the boxes = outliers. Blue lines represented the "zero value" lines.

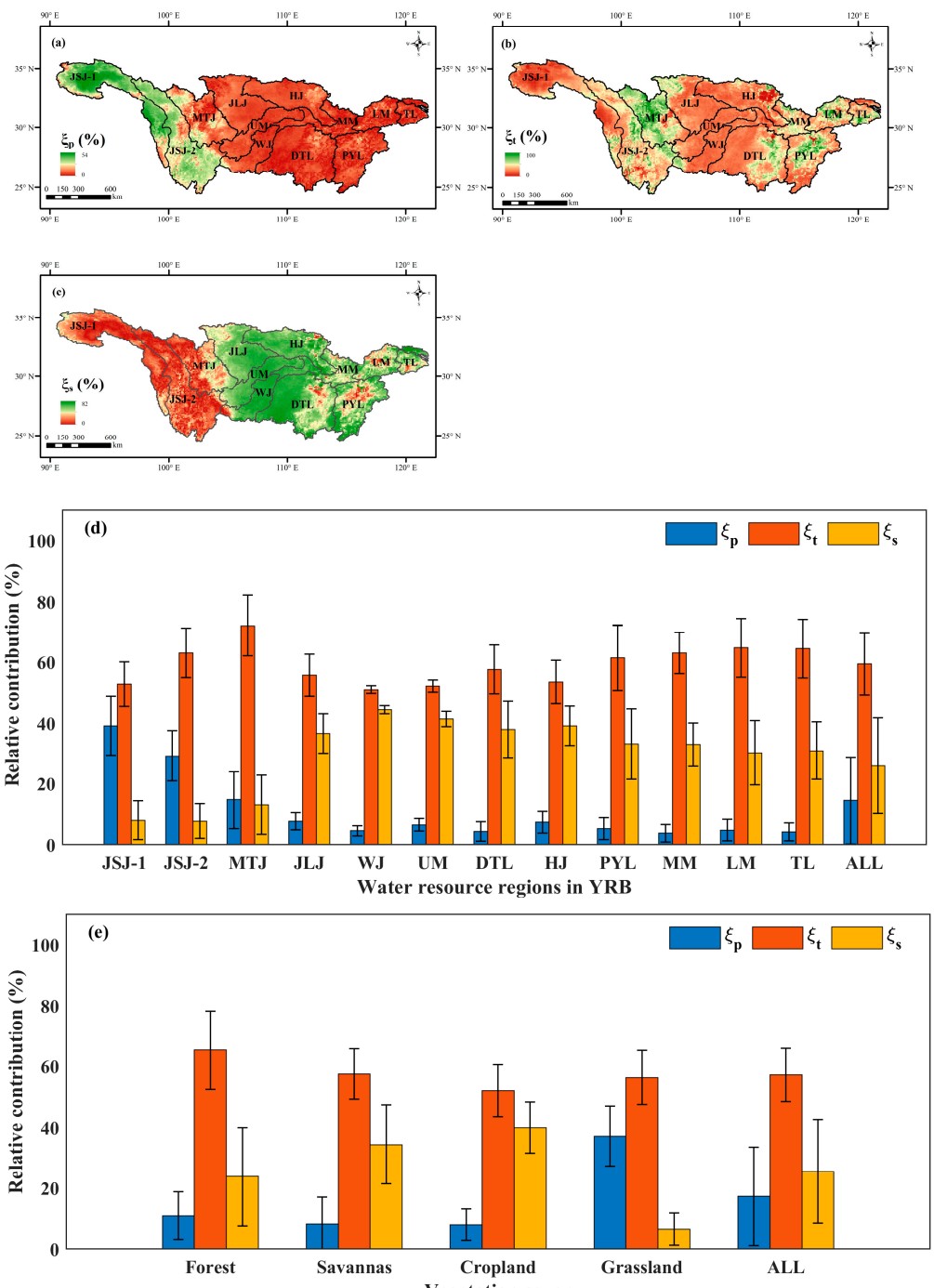

**Figure 7.** The spatial distributions of $\xi_p$ (**a**), $\xi_t$ (**b**) and $\xi_s$ (**c**) in the YRB. Histograms of $\xi_p$, $\xi_t$, and $\xi_s$ in 12 water resource regions (**d**) and vegetation types (**e**).

In the YRB, 72% of the areas had positive $\eta_p$ values, indicating that in most regions GPP increases with increasing precipitation (Figure 6a). The regions that exhibited greater sensitivities to precipitation were primarily located in the upper reaches of the Yangtze River, such as the Jinsha River Basin above Shigu (JSJ-1) and the Jinsha River Basin below Shigu (JSJ-2), with averages of 0.35 ± 0.07 and 0.29 ± 0.08, respectively. The relative contributions of precipitation to GPP in the Jinsha River Basin above Shigu (JSJ-1) and the Jinsha River Basin below Shigu (JSJ-2) regions were significantly higher than that in other water resource regions, with $\xi_p$ averages of 39.08 ± 9.68% and 29.12 ± 8.34% (Figure 7d). From vegetation type map (Figure 1e), it can be observed that these two water resource regions had significant proportions of grassland, which were 82.75% and 34.87%,

respectively. Our results also indicated that grassland was more sensitive to precipitation compared to other vegetation types (Figure 6e). Figure 7e also indicated grassland had much higher $\xi_p$ than other vegetation types, which was $37.12 \pm 9.83\%$.

Results showed that GPP of the 12 water resource regions and four vegetation types in the YRB were more sensitive to temperature changes (Figure 6b,d–e). The Mintuo River Basin (MTJ) and the Yangtze River mainstream below Hukou (LM) had much higher $\eta_t$ and $\xi_t$ values, with averages of $0.71 \pm 0.11$, $0.64 \pm 0.10$, $72.14 \pm 10.04\%$, and $62.09 \pm 6.81\%$, respectively (Figures 6d and 7d). $\eta_t$ of the Wu River Basin (WJ) and the Yibin-Yichang section of the Yangtze River mainstream (UM) were much lower compared to other water resources regions, with averages of $0.44 \pm 0.01$ and $0.41 \pm 0.03$. $\xi_t$ of those two regions were $52.17 \pm 2.00\%$ and $50.99 \pm 1.31\%$. In terms of vegetation cover, forests and savannas had much higher $\eta_t$ and $\xi_t$ values, which were $0.64 \pm 0.12$, $0.58 \pm 0.08$, $65.34 \pm 12.91\%$ and $57.47 \pm 8.23\%$, respectively (Figures 6e and 7e). Cropland had much lower $\eta_t$ and $\xi_t$ values with averages of $0.35 \pm 0.07$ and $51.98 \pm 8.50\%$.

## 4. Discussion

### 4.1. The Responses of GPP to the Climate Changes

Greening trends have been observed on both global and regional scales, based on emerging greening signals detected from satellites in the last decades. Earth System Modes (ESMs) also revealed a progressive increase in leaf area index (LAI) over the 21st century [49–51]. Climate changes predominantly affect ecosystems via alterations in temperature, precipitation, and shortwave radiation patterns. In this research, notable temperature increase was observed, while precipitation and shortwave radiation exhibited spatial heterogeneity in the YRB (Figures S1–S3), which was consistent with previous studies [44]. Warming climates may extend growing seasons and strengthen photosynthesis in plants, promoting better vegetation growth. Our results indicated forests were more sensitive to temperature changes. The YRB is in a typical subtropical monsoon climate with favorable moisture conditions that are sufficient for vegetation growth. Thus, the impact of changes in precipitation and shortwave radiation on GPP was relatively weak. However, our results also showed that, for the headwaters of the Yangtze River, the predominant vegetation type is grassland, which is vulnerable and particularly sensitive to climate change. The rise in temperature, coupled with reduced precipitation and radiation, can result in a decrease in GPP. Previous studies also found grasslands in cold regions may exhibit heightened sensitivity to climate changes [52]. This was probably attributed to their typically vast soil organic carbon reserves which are at risk of breaking down [53]. Furthermore, decomposition rates generally rise with increasing temperature [54].

### 4.2. Limitation and Uncertainty

This study focused on the responses of GPP in the YRB to the variations in climatic factors, and did not mention the GPP responses to anthropogenic activities. In recent years, the YRB had witnessed a series of ecological restoration initiatives, including programs like the Grain to Green Program and the Natural Forest Conservation Program, which played a significant role in enhancing China's forest coverage [55]. However, at the same time, urbanization (including population growth and economic development) can also exert negative impacts on vegetation growth. But we believed that this would not change the main conclusion of this study. As mentioned in the introduction, in this study, we only explored the impact of climate change on GPP. We applied PSLR method to explore the relationships between GPP and climate factors. The coefficients directly reflected the sensitivities of GPP to those three factors. However, it should be noted that the contribution analysis only reflected the "relative" contributions of those three factors and not their absolute contributions. Thus, we believed that the anthropogenic activities on GPP would not confound our conclusions. Our results showed that the lower reaches (i.e., the Taihu Lake Basin) had a decreasing GPP trend, which may be attributed to the rapid urbanization, as the Taihu Lake Basin stands in one of China's most prosperous areas,

located at the core region of the YRD. It is challenging to directly quantify the impacts of human activities on GPP. In future research, based on our understanding of the response pattern of GPP to climatic factors in this research, efforts can be made to quantify the influences of anthropogenic activities on GPP.

## 5. Conclusions

In this study, we explored the spatial and temporal variations of GPP and its responses to climate changes in the YRB from 2000 to 2018. The primary conclusions can be succinctly summarized as follows:

(1) The annual average GPP in the YRB was $1153.5 \pm 472.4$ g C m$^{-2}$ yr$^{-1}$ from 2000 to 2018. The GPP of the Han River Basin, the Yibin-Yichang section of the Yangtze River mainstream, and the Poyang Lake Basin were relatively high, while the GPP of the Jinsha River Basin above Shigu and the Taihu Lake Basin were relatively low.

(2) Significant increasing trends were observed in GPP over the 19-year period, with an annual increase rate of 8.86 g C m$^{-2}$ yr$^{-1}$ per year. The GPP of the Poyang Lake Basin and the Jialing River Basin grew much faster. Savannas and forests also had relatively higher GPP increasing rates. Greater emphasis should be placed on vegetation protection in the Taihu Lake Basin, as it had decreasing GPP trends.

(3) Temperature was the primary climatic driver of GPP changes in the YRB. The relative contributions of precipitation, temperature, and shortwave radiation to GPP variations in the YRB were $13.85 \pm 13.86\%$, $58.87 \pm 9.79\%$, and $27.07 \pm 15.92\%$, respectively. The regions with higher sensitivities to precipitation were primarily located in the upper reaches of the Yangtze River.

**Supplementary Materials:** The following supporting information can be downloaded at: https://www.mdpi.com/article/10.3390/f14091898/s1. Figure S1: The spatio–temporal changes of precipitation change rates (a) and significances (b) in the YRB; Figure S2: The spatio–temporal changes of temperature change rates (a) and significances (b) in the YRB; Figure S3: The spatio–temporal changes of shortwave radiation change rates (a) and significances (b) in the YRB.

**Author Contributions:** All authors contributed to the design of this research and the writing of the manuscript. Conceptualization, C.N. and Q.Y.; methodology, C.N.; software, Y.Z.; validation, R.X., C.D. and X.C.; writing—original draft preparation, C.N. and X.C. All authors have read and agreed to the published version of the manuscript.

**Funding:** This research was funded by the Fundamental Research Funds for the Central Public-Interest Scientific Institution, Chinese Research Academy of Environmental Sciences (Grant number 2022YSKY-72), the National Key Research and Development Program of China (Grant number 2022YFC3202104) and the Yangtze River Joint Research Phase II Program (Grant number 2022-LHYJ-02-0603).

**Data Availability Statement:** The data presented in this study are available on request from the corresponding author. The data are not publicly available due to privacy.

**Conflicts of Interest:** The authors declare no conflict of interest.

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
