# Peer review of "The Spatio-Temporal Variations of GPP and Its Climatic Driving Factors in the Yangtze River Basin during 2000–2018"

_forests, doi:10.3390/f14091898_

Round 1

Reviewer 1 Report

The paper entitled “The spatio-temporal variations of GPP and its climatic driving factors in Yangtze River Basin during 2000-2018” presents an interesting research which fits to the scope of the journal. However, it should be revised before publication. Below I present comments which should be considered while improving the paper.

1.      While introducing carbon dioxide circulation in the first section I think it would be valuable also to mention strategies which are applied to mitigate climate challenges like carbon sequestration. This process is commonly analyzed from the perspective of technological solutions, however, it can be achieved in more natural ways, including by wood production in forests. Please see for instance: Carbon Sequestration in Forest Valuation. Real Estate Management and Valuation, 2016, Vol. 24, No. 1, pp. 76-86

2.      Figure 1a – please verify the DEM used in the research. The legend presents that the range starts from -140 m. Moreover I suggest to replace units from “m” into “m asl”.

Reviewer 2 Report

The authors undergo an interesting study of the changes in GPP due to the temporal and spatial variability of temperature and precipitation in the Yangtze River basin. They presented evidences of the shape difference in GPP with latitude and longitude associated with differences in water availability and vegetation types as well as yearly GPP dynamics. They showed that different vegetation types also have different climate change responses.

I believe that the presented study can be published, however I would like to focus of the following parts that can be improved further:

In the Methods:  It would be good to provide an additional information on the spatial distributions of temperature and precipitation. The reported very wide spreads in values can be characteristics to both forests and savannas (grasslands) and should be further separated to make it clear. Additional figures could help.

In addition, looking at the Introduction and the literature sources, there is a predominant number of publications with the Chinese names. Because of it, I don't think the Introduction and the literature review is representative and must be improved for completeness. 

Kind regards,
